# Application of a Latent Transition Model to Estimate the Usual Prevalence of Dietary Patterns

**DOI:** 10.3390/nu13010133

**Published:** 2020-12-31

**Authors:** Andreia Oliveira, Carla Lopes, Duarte Torres, Elisabete Ramos, Milton Severo

**Affiliations:** 1EPIUnit—Instituto de Saúde Pública, Universidade do Porto, Rua das Taipas, 135, 4050-600 Porto, Portugal; carlal@med.up.pt (C.L.); dupamato@fcna.up.pt (D.T.); eliramos@med.up.pt (E.R.); milton@med.up.pt (M.S.); 2Department of Public Health and Forensic Sciences, and Medical Education, Faculty of Medicine, University of Porto, Rua Doutor Plácido da Costa, 4200-450 Porto, Portugal; 3Faculty of Nutrition and Food Sciences, University of Porto, Rua do Campo Alegre 823, 4150-180 Porto, Portugal

**Keywords:** feeding behavior, latent class analysis, latent transition analysis, usual intake, adults

## Abstract

Background: This study aims to derive habitual dietary patterns of the Portuguese adult population by applying two methodological approaches: a latent class model and a latent transition model. The novel application of the latent transition model allows us to determine the day-to-day variability of diet and to calculate the usual prevalence of dietary patterns. Methods: Participants are from the National Food, Nutrition and Physical Activity Survey of the Portuguese population, 2015–2016 (2029 women; 1820 men, aged ≥18 years). Diet was collected by two 24 h dietary recalls (8–15 days apart). Dietary patterns were derived by: (1) a latent class model using the arithmetic mean of food weigh intake, with concomitant variables (age and sex); (2) a latent transition model allowing the transition from one pattern to another, with the same concomitant variables. Results: Six dietary patterns were identified by a latent class model. By using a latent transition model, three dietary patterns were identified: “In-transition to Western” (higher red meat and alcohol intake; followed by middle-aged men), “Western” (higher meats/eggs and energy-dense foods intake; followed by younger men), and “Traditional-Healthier” (higher intake of fruit, vegetables and fish, characteristic of older women). Most individuals followed the same pattern on both days, but around 26% transited between “In-transition to Western” and “Western”. The prevalence of the dietary patterns using a single recall day (40%, 27%, 33%, respectively) is different from the usual prevalence obtained by the latent transition probabilities (48%, 36%, 16%). Conclusion: Three dietary patterns, largely dependent on age and sex, were identified for the Portuguese adult population: “In-transition to Western” (48%), “Western” (36%), and “Traditional-Healthier” (16%), but 26% were transient between patterns. Dietary patterns are, in general, deviating from traditional habits.

## 1. Introduction

Dietary patterns are valuable for capturing and summarizing the habitual dietary habits of a given population. They are particularly useful for examining the complex role of multiple dietary components in the etiology of diet-related chronic diseases [1,2,3]. Dietary patterns complement the study of the isolated effects of single foods or nutrients, which have limited applications for exploring the cumulative effects of diet in diseases for which multiple components are relevant. However, the identification of dietary patterns requires challenging methods of data collection and analyses. Usually, dietary patterns are defined using data from food frequency questionnaires (FFQs) [4,5,6], which are able to capture the usual food consumption. Nevertheless, other methods such as 24 h dietary recalls or food records have been used by averaging the food weight of a few dietary recalls, usually 2 or 3 days covering weekdays and a weekend day. European guidelines from national dietary surveys [7] support the collection of dietary data by using 24 h dietary recalls or food records, but it has inherent problems related with day-to-day variability of dietary intake [8,9], leading to zero inflations, or handling of dietary misreporting, which must be properly addressed.

Despite these methodological concerns, data-driven dietary patterns have been defined in several populations as an exploratory method and using different statistical techniques, such as principal component analysis, factor analysis and cluster analysis [10]. They allow for the aggregation of foods frequently consumed together or of individuals with similar food consumption. More recently, a few studies have used latent class analysis to model dietary transitions [11,12,13], but data often come from FFQs not requiring methods to deal with the day-to-day variability of dietary intake. By single 24 h recalls or food records, the more general approach to handle with the diet variability is to average the food weight in each day. This approach assumes that the mean of the food weigh intake of 2 or 3 days represents the usual intake of each individual. However, even after performing the average of several days of consumption, the within-person variability remains, and therefore, attenuating the true correlations between the food items. Appropriate methods to estimate the usual intake are required [14,15].

By using data from a nationwide Dietary Survey, this study applied a novel approach to derive habitual dietary patterns: a latent transition model that overcomes the day-to-day variability of dietary intake from two 24 h dietary recalls and allows the estimation of the usual prevalence of dietary patterns based on a transition probability matrix. It was compared with a latent class model that aggregates data by averaging dietary intake each day.

## 2. Materials and Methods

### 2.1. Participants

Participants were enrolled at the National Food, Nutrition and Physical Activity Survey of the Portuguese general population 2015–2016 (acronym: IAN-AF 2015–2016) [16,17]. It aimed to evaluate nationwide and regional dietary habits, physical activity and anthropometrics, and to evaluate their relationship with other health determinants.

A representative sample of the Portuguese general population, aged between 3 months and 84 years of age, was selected from the National Health Registry, by complex multistage sampling, in each of the seven Portuguese geographical regions (including Madeira and Azores islands).

Individuals living in collective residences or institutions, living in Portugal for less than 1 year (non-applicable to infants), non-Portuguese speakers, and with diminished physical and/or cognitive abilities that hamper participation were not included. A screening of cognitive impairment was performed in those aged 65 years or more at the beginning of interviews by using the Mini-Mental State Examination test [18]. Those with cognitive impairment, classified according to education level [19,20], only completed a physical examination.

### 2.2. Ethics

All participants were asked to provide their written informed consent according to the Ethical Principles for Medical Research involving human subjects expressed in the Declaration of Helsinki and the national legislation.

Ethical approval for the conduction of the Survey was obtained from the National Commission for Data Protection (authorization n° 4940/2015), the Ethical Committee of the Institute of Public Health of the University of Porto (authorization n° CE15033/2015), and the Ethical Commissions of each one of the Regional Administrations of Health from Portugal.

### 2.3. Sample Size Estimation

The sample size was estimated by assuming a mean population energy intake of 2000 kcal/day (standard deviation, SD = 500) and an effect size of 8% (a true difference of 40 kcal in the mean intake), with a confidence level of 95%. Considering the distribution of the Portuguese population and a design effect of 1.20 (an increase of 20% of the sample size), the sample size required to have representativeness at the national level was of 5068 individuals: 935 children and adolescents (0–17 years), 3262 adults (18–64 years), and 871 elderly subjects (65–84 years). More details about sample size estimation can be found in the research protocol publication [16].

During the field work, a total of 6553 participants completed only one of the two-day assessments; 5811 participants completed the two necessary interviews for dietary assessment; those aged 18–84 years (n = 3849; 2029 women, 1820 men) were included in the current study (Appendix A: Flowchart of participant’s selection). The participation rate among the eligible individuals was 29.6%. A comparison between those who participated and those who refused to participate showed that the latter were older and less educated (as expected in population-based surveys), although did not differ on the prevalence of fruit and vegetables consumption, practice of regular leisure-time physical activity and obesity [16]. In addition, the final sample was weighted according to sex and age groups to represent the number of individuals from the national general population [16].

### 2.4. Data Collection

The procedures for data collection were adapted from the European Food Safety Agency Guidance in view of the EU Menu methodology [7].

Two interviews were conducted by trained researchers with background in nutrition or dietetics, by using computer-assisted personal interviewing, for 12 months (from October 2015 to September 2016). The examination site was the participant’s home (<1%) or the Primary Health Care Unit they belong to, selected according to participant’s preference. The days of reporting were randomly allocated but were prone to change in case of unavailability of the participant in that day.

An electronic platform (You eAT & Move) was developed to assist data collection and the management of the field work. This e-platform includes the ”You” module to collect sociodemographic and other health-related data; the ”eAT24” module for collection of food consumption and dietary data by 24-h dietary recalls (or food diaries); and the “MOVE” module for data collection on physical activity.

#### Dietary Intake Assessment

Dietary intake was obtained by two non-consecutive 24 h dietary recalls, collected eight to 15 days apart. Interviews were distributed over the four seasons of the year and including all days of the week, in order to incorporate seasonal effects and day-to-day variability in dietary intake. On Mondays, 24 h recalls were randomly assigned to Saturdays and Sundays.

All foods, including beverages and dietary supplements consumed during a 24 h period, were recorded per eating occasion, based on an Automated Multiple-Pass Method for 24 h (5 steps) [21], reported in detail elsewhere [17], and described using the FoodEx2 classification system [22]. Foods and recipes were quantified as eaten by several methods, including an electronic picture book developed to be included in the ”eAT24” module [23]. Conversion of foods into nutrients was performed using by default the Portuguese food composition table [24], constantly adapted and updated during and after the field work.

### 2.5. Dietary Patterns’ Definition

Food groups were considered based on the European FoodEx2 classification system [22], and aimed to represent major contributors to the dietary habits of the Portuguese population. Those food groups that in 95% of the days had nil consumption were excluded from analysis (for instance, some alcoholic beverages, such as spirits). For the definition of dietary patterns, a total of 28 food groups were finally considered: dairy products (1 food group), meat and processed meats (3 food groups), fish (1 food group), eggs (1 food group), fats and oils (4 food groups), fruit and vegetables (5 food groups, including soups and natural juices), grains, cereals and tubers (3 food groups), confectionary (1 food group), salty snacks (1 food group), soft drinks (2 food groups), alcoholic beverages (2 food groups), water, teas and coffee (3 food groups), table sugar (1 food group), and artificial sweeteners (1 food group) (Appendix A: Food groups description). The intra-class correlation coefficient (ICC) for these food groups ranged from 0.11 to 0.79; 20% of food groups presented an ICC higher than 0.70 (Appendix A), but most of food groups presented an ICC between 0.30 and 0.70, meaning that these food groups have a within-person variability higher than 30%. When comparing data from the two 24 h recalls with a short FFQ for fruit and vegetables, also available for these participants, moderate Pearson correlation coefficients were found (fruits: rho = 0.54, 95% CI: 0.52–0.56; vegetables: rho = 0.33, 95% CI: 0.30–0.36; vegetable soup: rho = 0.41, 95% CI: 0.38–0.44). In addition, in a previous study, the accuracy of the eAT24 software to assess dietary intake was also measured against urinary biomarkers such as nitrogen, and potassium in a subsample of 94 subjects, suggesting that the software performed well in estimating protein and potassium intakes, by using two non-consecutive 24 h recalls [25].

To overcome outliers, non-continuous variables, and the usual inflation of zeros (i.e., a high percentage of individuals not consuming a particular food group, for example, more than 80% had a nil consumption of nectars, beer, and artificial sweeteners), each food group was divided into three categories: nil; equal or below the sample median and above the sample median intake (among the consumers). These food groups were treated as ordered variables. The food groups inclusion in each dietary pattern was based on the distribution of individuals across these three classes of food consumption: nil; equal or below the sample median and above the sample median intake. To define which food groups are more representative of each dietary pattern, the percentages of individuals in each class were observed. If they were very close, a food group could be “representative” for more than one dietary pattern.

Dietary patterns were driven using two methods: (a) latent class model using aggregated data (the arithmetic mean of food weigh intake in recall days); (b) latent transition model, without aggregating data. Both were driven using age and sex as concomitant variables; this means that the probabilities of latent class membership are conditional on age and sex. In a sensitivity analysis, dietary patterns were defined without considering age and sex, but models improved their fit with the inclusion of age and sex.

Ten to 100 random starting values for each latent class model were specified to obtain the final model. For all models, the number of extracted classes (patterns) and the different model parametrizations were decided based on the Bayesian Information criteria (BIC) and Akaike Information Criteria (AIC), the log-likelihood value, and relative entropy [26]. The entropy ranges between 0 and 1; a high value indicates a good classification.

Participants were classified into a class (dietary pattern), based on the highest probability of membership.

#### 2.5.1. Latent Class Model

The classification of each dietary pattern is estimated based on the mean of food weigh intake in the two days which is likely insufficient to remove the within-person variability of dietary intake. Therefore, the prevalence of dietary patterns is influenced by the within-person variability (which is not removed by averaging the food weight intake).

The latent class model assumes that the association among food items is explained by the latent class variable (with K categories). The interpretation of the latent class model is based on the distribution of the food items in each category conditional on the class membership. Several models, specifying between one and seven latent classes, were tested.

#### 2.5.2. Latent Transition Model

In the latent transition model, it is assumed that the subject may change from one pattern to another. The major difference between a latent class model and a latent transition model is that the probability of transition to the same or different latent status at time t + 1 is conditional on the latent status at time t. This is performed by multinomial regression where t + 1 latent categorical variable is regressed on time t latent categorical variable [27]. Thus, the latent transition model has the same assumption as the typical first order Markov Chain model.

Thus, the usual prevalence of dietary patterns was calculated by using the transition probabilities obtained.

The latent transition model was applied considering two time points: day 1 and day 2 of dietary intake, each with two to three latent classes in order to identify the best model.

Considering that the two days of recall were apart between eight to 15 days, we assumed measurement invariance across time, meaning that the dietary patterns were measured the same way on both days. In addition, we tested a model constraining the effect of the concomitant variables to be the same in both days. These two assumptions were tested previously by constrained and non-constrained models, but they improved their fit with the constrained version.

#### 2.5.3. Usual Prevalence of Dietary Patterns

*C*_*d*−1_ Considering that a latent class variable at day *d*, *C*_*d*_ has K classes, we assumed that the transitions probabilities from day *d* − 1 to day d only depend on the state *C*_*d*−1_, i.e., depend on the previous state and not on the preceding steps (first order Markov Chain model assumption). So, the transition probabilities for *d* = 2 (days) and k = 3 (classes) are represented by the following matrix:(1)$$P=\left[\begin{array}{ccc} P\left(C_{2}=1|C_{1}=1\right) & P\left(C_{2}=2|C_{1}=1\right) & P\left(C_{2}=3|C_{1}=1\right)\\ P\left(C_{2}=1|C_{1}=2\right) & P\left(C_{2}=2|C_{1}=2\right) & P\left(C_{2}=3|C_{1}=2\right)\\ P\left(C_{2}=1|C_{1}=3\right) & P\left(C_{2}=2|C_{1}=3\right) & P\left(C_{2}=3|C_{1}=3\right) \end{array}\right]\;S_{1}$$

*S*_1_ = [*π*_1_   *π*_2_   *π*_3_]*S*_1_ is called the initial state matrix, with number of entries equal to the number of latent classes, K, and has the probability (*π*) of being in a specific class in day 1.

*S*_∞_ However, we wanted to estimate the probabilities at long-run (state prevalence), *S*_∞_, where the number of days, *d*, goes to infinity. This is only possible if $S_{\infty }$ converged for a fixed value. If P matrix is regular, i.e., all entries are positive (or at least for one of $P^{d-1}$), then $S_{\infty }$ is stationary.

Considering state probabilities at day 2, 3,…, d, i.e., $S_{2}$, $S_{3}$,… $S_{d}$, it can be calculated as:(2)$$S_{2}=S_{1}\times P,\;S_{3}=S_{2}\times P=S_{1}\times P^{3-1},\ldots ,\;S_{d}=S_{1}\times P^{d-1}$$

To calculate the state prevalence of each dietary pattern, we used the previous formula to obtain the *S*_∞_, if *P* is a regular matrix.

In the final latent transition model, the transition probabilities estimated were also conditional on age and sex.

From now on, state prevalence will be named as usual prevalence.

### 2.6. Dietary Pattern Description

For dietary pattern description (and not for their definition), all prevalence estimates, and the respective 95% confidence intervals (95% CI) were weighted according to the complex sampling design, considering stratification by the seven Portuguese geographical regions (NUTS II) and cluster effect for the selected Primary Health Care Unit.

Prevalence and mean estimates according to nutritional intake, geographical region, socioeconomic and health status were presented with standardization for age and sex across dietary patterns. Prevalence estimates were compared by the Rao-Scott adjusted chi-square statistic. Significance was based on the adjusted F and its degrees of freedom. Weighted means (95% CI) were compared by test of Model effects (Wald F test) and presented with standardization for age and sex across dietary patterns, as well.

The latent models were fitted by using MPlus^®^ (version 5.2; Muthem & Muthem, Los-Angeles, CA, USA) [28]. Further analyses were performed using the library “survey” of R^®^ software (The R Project for Statistical Computing), version 3.4.0 for Windows. A significance level of 5% was assumed.

## 3. Results

Table 1 presents the sample size and the percentage of membership to each dietary pattern identified by the two methods: latent class model and latent transition model, both with age and sex as concomitant variables.

By using a latent class model with no concomitant variables, five dietary patterns (DPs) were identified (AIC: 179,357.7; BIC: 181,134 and entropy: 0.678). By including age and sex as concomitant variables, the model had a better fit (AIC: 176,843.2; BIC: 179,039 and entropy: 0.763) and identified six DPs, which were dependent on age and sex of the Portuguese population.

Considering the six classes of the latent class model (six DPs), the expected number of classes in the latent transition model would be between 2 or 3 classes considering that this corresponds to 2 by 2 combinations or 3 by 3 combinations in the two reporting days.

The two-classes of the latent transition model with concomitant variables (age and sex) had AIC, BIC, and entropy of 326,573.268, 327,305.169, and 0.961, respectively. The three classes without concomitant variables had AIC, BIC, entropy values of 324,371.168, 325,472.148, and 0.902, respectively. The inclusion of the concomitant variables improved the fit indices (AIC: 323,542.011; BIC: 324,655, and Entropy: 0.893) compared to the two previous latent transition models, being considered the best model.

Figure 1 provides a schematic description of the profile of each DP identified based on the highest and lowest intake of major food groups. Dietary patterns identified by the latent class model are somewhat comparable to the three DPs derived by the latent transition model. DP4 and DP5 from the latent class model are comparable to “In-transition to Western”; DP1 and DP2 are comparable to “Western” and “Traditional-Healthier”; and DP6 corresponds to the “Traditional-Healthier”. A detailed description of the predicted class membership to define dietary patterns derived by the latent class model is presented in Appendix A (Appendix A: Probabilities of food group consumption conditional on a dietary pattern derived by the latent class model with concomitant variables (age and sex)).

By using the latent transition model, three DP were identified. DP1 has a higher percentage of individuals in the upper category of intake (above the sample median) of legumes; starchy foods, such as pasta, rice, potatoes and bread; red and processed meats; and wine, but also of sweets, sugar, soft drinks and coffee (referred to from now on as “In-transition to Western”) (Table 2). DP2 is characterized by those having the highest intake of vegetables, protein foods, such as all type of meats and eggs, starchy foods (pasta/rice/potatoes), salty snacks, sweets, nectars/soft drinks, and water compared with the other DP (referred to from now on as “Western”). DP3 has a higher percentage of individuals with the highest intakes (in the upper category) of vegetables, vegetable soup, fruits, dairy, fishery, olive oil, sweeteners and teas and the lowest intake of beer, red and processed meat, salty snacks, and soft drinks (referred to as “Traditional-Healthier”) (Table 2).

According to the latent transition probabilities of the estimated model, most individuals follow the same DP in both days (88.4% for DP1, 85.4% for DP2, and 97.1% for DP3). However, 11.4% transit from “In-transition to Western” to “Western” and 14.6% the other way around. Almost no one moves from “In-transition to Western” or “Western” to the “Traditional-Healthier” DP (Figure 1b).

Figure 2 shows the transitional probabilities by age and sex based on the estimated latent transition model for each DP. It shows that there is an increasing transition with age from “In-transition to Western” to the others (mainly “Western”). Transition from “Western” to “In-transition to Western” decreases with age and is very similar by sex. There is an increasing transition from the “Traditional-Healthier” DP to the others (mainly “Western”) with increasing age, and a clear sex effect (transition is faster in women).

Age is negatively associated with “In-transition to Western” (OR = 0.949 95%CI: 0.934 to 0.964) and “Western” (OR = 0.920, 95% CI: 0.908 to 0.933) compared to “Traditional-Healthier”. Males report more frequently “In-transition to Western” (OR = 6.13, 95% CI: 2.549 to 14.939) and “Western” (OR = 6.17, 95% CI: 2.762 to 13.622) DP compared to the “Traditional-Healthier”.

Table 3 provides the description of the three groups of individuals identified by the latent transition model according to nutritional intake, geographical region, socioeconomic, and health status. Estimates are presented with standardization by age and sex across DP. Standardization allows estimating the associations independently of the age and sex distribution of each DP (which will have a major influence in the reported associations). As shown in Table 1, there are large differences in sex and age across DP.

The highest mean intake of energy, carbohydrates, including mono and disaccharides, alcohol, and sodium is among the followers of the “In-transition to Western” DP. Individuals classified as having the “Western” DP have the highest mean intake of protein and fats, including trans-fatty acids. The “Traditional-Healthier” DP has the highest mean values of fiber, calcium, and vitamin C intake.

After standardization by age and sex across DP, no geographical disparities were observed (Table 3). Followers of the “In-transition to Western” are less educated, and more often under/normal weighted. The “Western” DP is more often reported by more educated subjects, more frequently with obesity, less physically active, but also with lower prevalence of diseases needing medical care. The “Traditional-Healthier” followers are the most educated and physically active, but have a higher prevalence of pre-obesity and obesity and more diseases needing medical care (Table 3).

The prevalence of each dietary pattern by using a single recall day, corresponding to matrix S1 (initial state matrix) and S2 (day 1 and day 2, respectively) or the usual prevalence obtained by the latent transition model is described in Figure 3. The classification into the three DPs is very similar considering the first or the second day of recall (DP1~40%; DP2~27%; DP3~33%), but assuming the usual intake, 48% of the Portuguese follows “In-transition to Western” (DP1), 36% follows “Western” (DP2) and only 16% follows the “Traditional-Healthier” DP. Previous estimates, not considering usual intake, will wrongly estimate this usual prevalence.

## 4. Discussion

To the best of our knowledge, no previous study has applied a latent transition model to derive the habitual DP of a given population, by using dietary intake from 24 h recalls. This method allows taking into account the within-person variability of dietary intake, typical of multiple short-term recalls or records, and to estimate the usual prevalence of DP. It identified three DPs: “In-transition to Western”, “Western”, and “Traditional-Healthier”, but five groups of subjects. Two of these (~26%) have a usual intake that transits between DPs.

It is important to note that most individuals follow the same DP in both recall days, but around 26% transit from “In-transition to Western” to “Western” or the reverse. Less than 1% transit to the “Traditional-Healthier” DP; this means that those following the “In-transition to Western” or the “Western” never follow the “Traditional-Healthier”, as well as the reverse.

By using the latent transition model, it was possible to estimate the usual prevalence of DP considering the usual intake (at the long-term). The latent class analysis corrects for the within-person variability by averaging the 2 days of dietary intake that are already known as insufficient (as shown by the ICC < 0.70 for most food groups). The latent transition model overcomes this by using the transition probabilities matrix. The initially observed percentages of membership were 40% (DP1), 27% (DP2), and 33% (DP3). After correcting for the within-person variability of dietary intake, the percentages were 48% (DP1), 36% (DP2), and 16% (DP3). This decreasing prevalence of the “Traditional-Healthier” DP can be explained by the low transition probabilities from the other DP. In addition, although of low magnitude, around 3% transit from “Traditional-Healthier” to “Western”. This group of older women may include two types of individuals: the healthier and the under reporters, suggested by the lower caloric intake and higher prevalence of obesity among them. Thus, we believe that the transition probabilities may correct this prevalence, because some of the under reporters are only under-reporters in one day, and so they will transit to another DP; the true usual prevalence of the “Traditional-Heathier” dietary pattern should be close to 16%.

Dietary patterns were highly associated with sex and age, so considering models conditional on age and sex improves the model’s fit. The “In-transition to Western” is more frequently followed by middle-aged men. In the future, it is important to understand if there is a cohort effect (those who were born in the 70 s have already changed their food habits to more westernized ones). During the 70 s, Portugal changed from a dictatorial, conservative political regime to a more independent republican state, justifying in part a deviation from more traditional habits by an entire generation. The “Traditional-Healthier” is more frequently reported by older women (those born in the 60 s, when the country had a climate of intense political instability and food paucity). The “Western” DP is followed by younger individuals, for whom access to more energy-dense foods became available and attractive. If we assume that food habits tend to track over time [29,30], a cohort effect rather than an age effect may be present in our results. Accordingly, it is not expected that the population change their food habits throughout life, i.e., that in older ages, they will adopt healthier DP; it is rather expected that in a few decades, DP will worsen with health-related implications. Additionally, a higher educational level was associated with the “Traditional-Healthier” DP. As previously suggested, a healthier diet in individuals from high socio-economic status is among the mechanisms that explains social differences in health [31]. Indeed, in high-income countries, high-socioeconomic status individuals are more likely to consume healthy foods such as whole grains, lean meats, fish, and fruit and vegetables, whereas more disadvantage socioeconomic individuals tend to consume more fat and less fiber [31,32]. Thus, younger and less educated subjects should receive a special focus when designing health promotion strategies related with diet.

The DP identified for the first time in Portugal by using a national representative sample represents a transition of more traditional and healthier DP to eating habits with a strong westernization, including foods such as salty and sweet snacks and soft drinks. Nonetheless, Southern Europe has been described as having healthier DP compared with northern countries, namely, the Mediterranean diet and the Atlantic Diet, with well-recognized health benefits [33,34,35].

In the present study, the DP called “In-transition to Western” had the highest percentages of membership. This pattern has similarities with the Atlantic Diet that has been defined as the traditional diet from Northern Portugal and Galiza, Spain [36]. It differs from the Mediterranean diet by a higher consumption of meats, particularly red meat; fishery; dairy; cereals, particularly potatoes and non-refined bread; and cooked vegetables (soups), and shares with the Mediterranean diet a moderate consumption of wine [35,36]. In the current study, the “In-transition to Western” has a high percentage of individuals with the highest intake (in the upper category of intake) of red meat and processed meat; starchy foods (pasta/rice/potatoes, bread); alcoholic beverages, particularly wine, but also sweets, soft drinks, and coffee. Thus, it includes more traditional foods, but also includes a high intake of energy-dense, poor-micronutrient foods as well. Although the “traditional” Atlantic Diet has been shown to be protective against myocardial infarction and cardio-metabolic health [34,35], the health impact of these important changes in traditional habits should be addressed in future studies.

The second group of individuals is characterized by the highest intake of protein and energy-dense foods (followed more frequently by younger men). It also shows a “westernization” process of food habits. These two DPs include a higher percentage of individuals with the highest intake of industrial foods, such as salty snacks, sweets, and soft drinks, suggesting that traditional food habits are in transition to more westernized ones. A previous study, using data from household food availability from 1995 to 2005 (more than 10 years ago) has already concluded that Portugal is gradually moving away from the traditional Mediterranean diet [37]. Lifestyles changes, food globalization, economic, and socio-cultural factors are being described as responsible for the decreasing adherence to the Mediterranean diet [38].

Nonetheless, according to our data, it is also true that some individuals (16%) are characterized by a higher intake of foods, such as vegetables, including vegetable soup; fruits; dairy; fishery; and olive oil. This “Traditional-Healthier” pattern is closer to the Mediterranean Diet, and more frequently reported by older women. Curiously, it was more frequently followed by overweight subjects (they could be already dieting or could be under reporters) and with diseases needing medical care, highlighting a possible reverse causality bias, due to the cross-sectional nature of this study.

An important methodological limitation of previous methods to derive DP, such as principal component analysis, is the use of food groups as continuous variables. In fact, the existence of zero inflations frequently occurs in dietary data, i.e., a high percentage of individuals not consuming a particular food group, especially if a single recall day is used. For example, in our study, legumes, beer, eggs, margarines, juices, and teas showed more than 80% of subjects with daily nil consumption in one of the reported days. For these food groups, the association/correlation will be attenuated; thus, the number of DP identified will be fewer and more inaccurate. Thus, they must not be treated as continuous variables. To overcome this methodological constrain, in our study, three classes of food consumption were considered: nil, below or equal the median, and above the median intake, and the predicted membership to one of these classes was estimated. This procedure was also important to minimize the impact of outliers of dietary intake. In our case, outliers were classified in the lower or upper classes, and are unlikely to influence the distribution of the variable. On the other hand, the choice of three categories could limit the variability of distribution.

Another concern is related to the external validity of the current results due to the relatively low participation rate among the eligible (around 30%). However, considering the burden of a dietary survey, including two interviews, this participation rate is acceptable and similar to other European surveys that used similar sampling approaches [17]. Moreover, participants in the survey were older and less educated than those who refused to participate, but no differences were found regarding key exposures, such as prevalence of fruit and vegetable intake, practice of regular leisure-time physical activity, and obesity status. In addition, the participant’s characteristics (sex and age) were compared with the information from CENSUS-National Institute of Statistics (INE), and after weighting, the final sample of this Survey shows a similar distribution of the population living in Portugal [17]. Lastly, we only used two days to report usual dietary intake, 8–15 days apart, and thus for those who have transited from one dietary pattern to another, we cannot accurately estimate which is the actual dietary pattern. In addition, we did not consider in analysis weekday versus weekend days. Nonetheless, the reported days were randomly assigned across seven days of the week and across 12 months of the year, minimizing variability concerns.

## 5. Conclusions

Dietary patterns are largely dependent on age and sex; they should be derived taking into account these concomitant variables. The latent transition model allowed us to identify only three DPs, but five groups of subjects, two of them whose usual intake transit between DPs (11.4% from “In-transition to Western” to “Western” and 14.6% in the reverse direction). Considering these high transition probabilities, the usual prevalence of each DP is considerably different (48% DP1, 36% DP2, 16% DP3) from those considering estimations based on a single recall day (40% DP1, 27% DP2, 33% DP3). The methodology used to derive DP allowed us to minimize the within-person variability of dietary intake from few days of report, which was often discarded in previous studies.

The heterogeneity of country-specific DP and their deviation from traditional habits will potentially enhance evidenced-based decisions for policy planning and management of programs related to the improvement of nutritional status.

## Figures and Tables

**Figure 1 nutrients-13-00133-f001:**
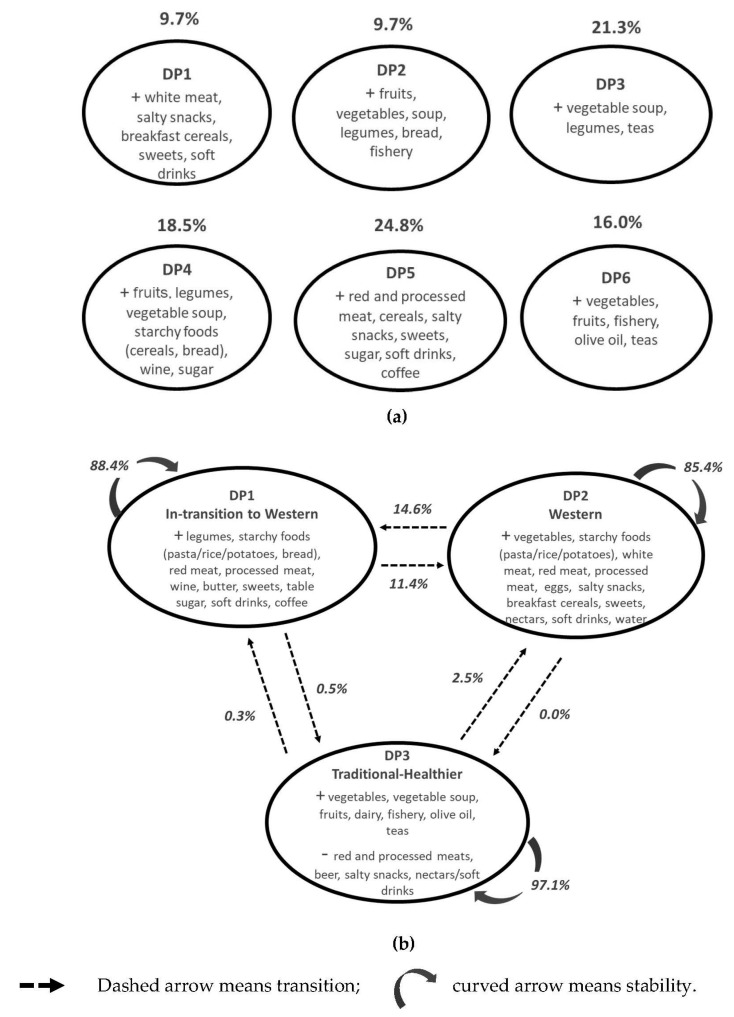
Description of dietary patterns: (**a**) latent class model (with concomitant variables); (**b**) latent transition model (with concomitant variables). Percentages (%) are the latent class probabilities (1a) or the latent transition probabilities (1b) based on the estimated model; food group descriptions are made based on the highest (+) and lowest intake (-) of major food groups.

**Figure 2 nutrients-13-00133-f002:**
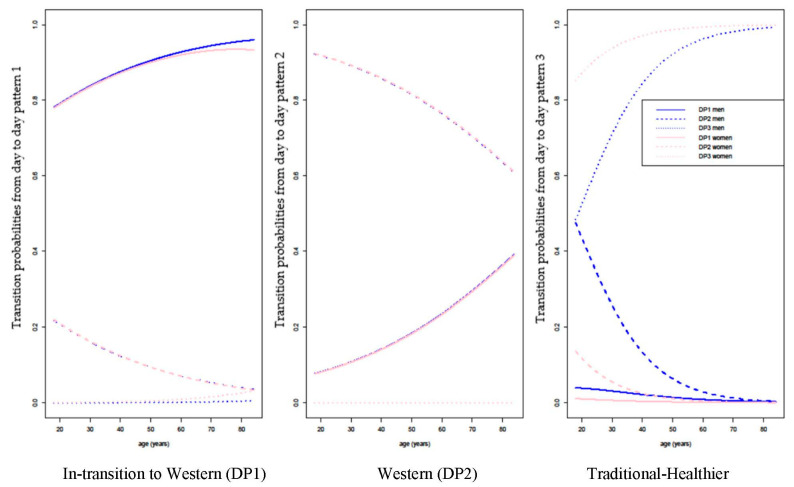
Transitional probabilities by age and sex based on the estimated model (latent transition model) for each dietary pattern. DP: dietary pattern.

**Figure 3 nutrients-13-00133-f003:**
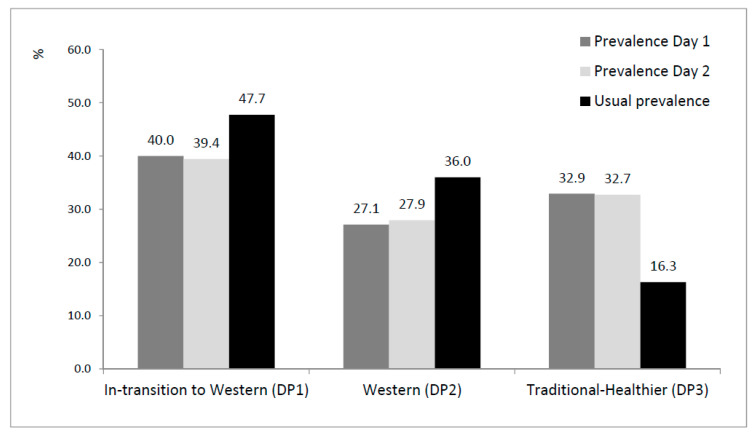
Prevalence of each dietary pattern considering a single recall day (Day 1 or Day 2) and usual prevalence obtained by the latent transition model (without sampling weighting). DP: dietary pattern.

**Table 1 nutrients-13-00133-t001:** Sample size and estimated percentage of membership in each dietary pattern identified by the two methods: Latent class model and Latent transition model, both with age and sex as concomitant variables.

	Sample Size (n = 3849)	$\mathrm{Estimated}\;\mathrm{Population}\;(\hat{N})$	Estimated Membership % (95% CI)	% Women (95% CI)	Mean Age, Years (95% CI)
1—Latent class model					
DP1	368	892,374	9.7 (8.4–11.2)	52.3 (45.4–59.0)	25.9 (24.7–27.0)
DP2	412	890,638	9.7 (8.4–11.1)	6.0 (2.7–12.6)	57.5 (55.8–59.2)
DP3	840	1,961,101	21.3 (19.3–23.6)	94.3 (91.1–96.4)	56.1 (54.5–57.8)
DP4	611	1,696,176	18.5 (16.3–20.8)	8.1 (4.1–15.6)	59.0 (57.9–60.1)
DP5	964	2,278,374	24.8 (22.6–27.1)	37.9 (33.1–43.0)	37.0 (35.8–38.1)
DP6	654	1,469,998	16.0 (14.0–18.2)	93.2 (90.2–95.3)	49.3 (47.4–51.1)
2—Latent transition model					
DP1	1411	3,632,681	39.5 (37.1–42.1)	38.6 (35.1–42.3)	47.5 (46.3–48.6)
DP2	1071	2,339,378	25.5 (23.4–27.7)	39.5 (35.1–44.2)	36.3 (34.8–37.8)
DP3	1367	3,216,602	35.0 (32.8–37.3)	75.0 (72.4–77.3)	57.1 (55.9–58.3)

DP: Dietary pattern; CI: confidence interval.

**Table 2 nutrients-13-00133-t002:** Probabilities of food group consumption ^1^ conditional on a dietary pattern derived by the latent transition model with concomitant variables (age and sex) ^2^.

		Nil	≤Median	>Median		Nil	≤Median	>Median
DP1	vegetables	27.2%	39.0%	33.8%	olive oil	82.8%	12.8%	4.4%
DP2		27.5%	33.1%	**39.4%**		85.7%	10.5%	3.8%
DP3		22.4%	38.3%	**39.3%**		73.7%	18.9%	**7.5%**
DP1	fruits	**30.5%**	37.8%	31.7%	butter	68.7%	20.2%	**11.1%**
DP2		**31.9%**	36.1%	32.0%		75.1%	17.9%	7.0%
DP3		8.4%	41.9%	**49.6%**		76.6%	17.8%	5.5%
DP1	legumes	84.7%	7.3%	**8.0%**	margarines	82.4%	12.0%	5.6%
DP2		88.3%	5.4%	6.3%		87.2%	8.4%	4.4%
DP3		87.3%	7.1%	5.6%		76.4%	18.3%	5.3%
DP1	vegetable soup	**62.3%**	18.6%	19.1%	sweets	0.0%	52.0%	**48.0%**
DP2		**67.3%**	16.7%	16.0%		34.9%	18.5%	**46.6%**
DP3		38.2%	32.8%	**29.0%**		23.2%	47.6%	29.2%
DP1	dairy	18.7%	41.5%	39.9%	table sugar	0.4%	46.2%	**53.4%**
DP2		19.8%	40.2%	40.0%		**97.6%**	1.6%	0.8%
DP3		13.6%	42.2%	**44.2%**		67.9%	23.5%	8.6%
DP1	cereals and tubers	14.3%	36.7%	**49.0%**	artificial sweeteners	98.3%	1.1%	0.5%
DP2		18.9%	37.5%	**43.6%**		95.1%	1.8%	3.1%
DP3		26.8%	46.3%	26.9%		85.1%	8.2%	6.7%
DP1	bread	10.2%	41.9%	**47.8%**	wine	62.6%	15.4%	**22.0%**
DP2		19.0%	41.1%	39.9%		74.4%	12.5%	13.1%
DP3		8.3%	48.9%	42.8%		73.2%	17.6%	9.2%
DP1	breakfast cereals	87.9%	5.8%	6.3%	beer	85.8%	7.6%	6.5%
DP2		76.2%	9.8%	**14.0%**		88.8%	5.5%	5.7%
DP3		83.0%	11,0%	5.9%		**98.2%**	1.6%	0.2%
DP1	white meat	65.4%	18.6%	16.0%	nectars	90.3%	4.9%	4.8%
DP2		62.0%	18.2%	**19.9%**		85.4%	7.8%	**6.8%**
DP3		70.2%	20.8%	9.0%		**96.6%**	2.7%	0.7%
DP1	red meat	46.8%	24.6%	**28.6%**	soft drinks	67.9%	15.9%	**16.2%**
DP2		52.8%	19.7%	**27.5%**		67.5%	14.7%	**17.8%**
DP3		**66.1%**	22.5%	11.5%		**94.1%**	4.8%	1.1%
DP1	processed meat	62.4%	17.6%	**20.0%**	natural fruit juices	95.3%	1.7%	3.0%
DP2		62.8%	18.4%	**18.7%**		94.0%	2.0%	4.0%
DP3		**80.6%**	13.8%	5.6%		90.5%	6.5%	3.1%
DP1	fishery	58.2%	20.9%	20.9%	coffee	1.9%	47.3%	**50.8%**
DP2		58.8%	18.5%	22.8%		54.5%	19.6%	26.0%
DP3		51.8%	25.9%	**22.4%**		19.2%	45.1%	35.7%
DP1	eggs	83.0%	9.7%	7.3%	teas	85.7%	10.3%	3.9%
DP2		81.7%	8.3%	**10.1%**		87.5%	7.1%	5.5%
DP3		83.4%	8.8%	7.8%		60.2%	24.6%	**15.2%**
DP1	salty snacks	84.7%	7.8%	7.5%	water	12.5%	48.4%	39.1%
DP2		80.9%	9.2%	**9.9%**		11.3%	38.0%	**50.7%**
DP3		**93.3%**	3.5%	3.2%		8.0%	48.3%	43.7%

^1^ Food group consumption divided into three categories: nil intake, equal/below and above median intake. ^2^ DP1—39.5%; DP2—25.5%; DP3—35.0% (BIC: 324,655). Major differences are in bold-type.

**Table 3 nutrients-13-00133-t003:** Dietary patterns (DPs) derived by the latent transition model with age and sex as concomitant variables, according to nutritional intake, geographical region, socioeconomic, and health status.

	Weighted Mean/%, Standardized for Age and Sex, 95% CI ^1^			
	DP1—In-Transition to Western 39.5%	DP2—Western 25.5%	DP3—Traditional-Healthier 35.0%	*p*-Value
Nutritional intake, mean				
Energy (kcal/day)	**2057 (2009–2104)**	1962 (1901–2022)	1686 (1621–1752)	<**0.001**
Protein (g/day)	87.7 (85.4–90.1)	**89.1 (86.2–92.1)**	75.6 (72.3–78.8)	<**0.001**
Carbohydrates (g/day)	**234.0 (227.3–240.7)**	205.6 (199.0–212.2)	191.9 (183.2–200.5)	<**0.001**
Mono and disaccharides	**90.9 (88.3–93.5)**	76.9 (73.2–80.7)	74.9 (71.7–78.0)	<**0.001**
Fat (g/day)	67.3 (65.6–69.1)	**68.1 (65.0–71.1)**	55.6 (52.9–58.4)	<**0.001**
Saturated fat	21.8 (21.1–22.5)	**21.9 (20.7–23.0)**	17.2 (16.2–18.3)	<**0.001**
Monounsaturated fat	27.6 (26.8–28.4)	**28.0 (26.5–2.4)**	22.8 (21.7–23.9)	<**0.001**
Polyunsaturated fat	10.8 (10.4–11.1)	**11.5 (11.0–12.0)**	9.5 (9.1–10.0)	<**0.001**
Trans fatty acids	0.85 (0.80–0.90)	**0.88 (0.81–0.95)**	0.68 (0.62–0.74)	<**0.001**
Alcohol (g/day)	**12.9 (11.6–14.2)**	**13.3 (11.6–15.1)**	6.8 (5.8–7.8)	<**0.001**
Fiber (g/day)	17.7 (17.3–18.2)	17.1 (16.4–17.8)	**19.2 (18.4–20.1)**	**0.001**
Calcium (mg/day)	728.4 (702.8–753.9)	731.3 (690.5–772.1)	**810.1 (767.5–852.7)**	**0.003**
Vitamin C (mg/day)	105.2 (98.1–112.3)	125.1 (109.3–14.09)	**132.4 (121.7–143.1)**	**0.005**
Sodium (mg/day)	**3294.1 (3195.9–3392.4)**	3110.9 (2977.1–3244.6)	2843.7 (2692.6–2994.9)	<**0.001**
Geographical region (NUTS II), %				
North	42.1 (38.7–45.4)	20.4 (17.1–23.8)	37.5 (34.1–40.8)	0.140
Center	41.3 (36.3–46.4)	24.1 (19.3–28.9)	34.5 (28.3–40.7)	
Lisbon Metropolitan Area	40.4 (35.6–45.2)	29.2 (24.7–33.7)	30.4 (25.6–35.2)	
Alentejo	37.2 (30.9–43.6)	25.6 (17.5–33.7)	37.2 (29.7–44.6)	
Algarve	31.1 (25.0–37.2)	34.6 (27.2–41.9)	34.3 (26.0–42.7)	
Madeira	35.5 (26.0–54.0)	28.9 (21.6–36.1)	35.6 (24.1–47.1)	
Azores	36.6 (26.2–46.9)	36.2 (25.2–47.1)	27.2 (19.5–34.9)	
Educational level, %				
None,1st and 2nd cycle	**42.8 (40.0–45.5)**	25.6 (22.8–28.4)	31.6 (28.4–34.8)	<**0.001**
3rd cycle and high school	**43.9 (41.3–46.5)**	23.3 (21.1–25.6)	32.8 (30.1–35.4)	
Higher education	30.5 (26.7–34.3)	**27.4 (23.2–31.5)**	**42.1 (37.7–46.6)**	
Body mass index categories ^2^, %				
Underweight/normal weight	**45.2 (42.2–48.1)**	23.9 (21.3–26.4)	31.0 (27.8–34.1)	**0.004**
Pre-obesity	37.5 (35.0–40.0)	23.5 (20.9–26.2)	**38.9 (36.4–41.5)**	
Obesity	39.4 (35.7–43.1)	**26.4 (23.1–29.7)**	**34.1 (29.6–38.6)**	
Perceived health status, %				
Excellent/Good	40.6 (38.9–42.3)	27.0 (25.1–28.9)	32.4 (30.4–34.3)	**0.081**
Reasonable	41.1 (38.4–43.7)	22.1 (19.5–24.7)	36.8 (33.8–39.9)	
Poor/Fair	38.4 (31.3–45.5)	23.9 (16.1–31.7)	37.7 (31.8–43.6)	
Disease with medical care, Yes, %	36.2 (33.4–39.0)	26.1 (23.8–28.4)	**37.7 (35.6–39.8)**	**0.005**
Regular sports practice, Yes, %	35.2 (32.0–38.4)	27.5 (24.7–30.4)	**37.2 (34.7–39.8)**	**0.005**

CI: confidence interval; DP—Dietary pattern; NUTS: Nomenclature of Territorial Units for Statistical Purposes. ^1^ Mean/Prevalence (%) weighted for the distribution of the Portuguese population, and standardized for age and sex across dietary patterns, and the respective 95% confidence interval. Major differences are in bold-type. ^2^ Body mass index categories defined according to the World Health Organization criteria: <25.0 kg/m^2^ (Underweight/normal weight); 25.0–29.9 kg/m^2^ (Pre-obesity); ≥30.0 kg/m^2^ (Obesity).

## Data Availability

Data is available at https://ian-af.up.pt/.

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
