# Peer review of "Application of a Latent Transition Model to Estimate the Usual Prevalence of Dietary Patterns"

_nutrients, 2020, doi:10.3390/nu13010133_

Round 1

Reviewer 1 Report

This is an interesting study with good scientific quality and good presentation.

The paper needs minor language editing (Language/Syntax issues).

Author Response

The paper was extensively revised and language was checked. If you fell we need a more professional language editing, we are willing to provide it.

Reviewer 2 Report

Dear Authors,

I read the manuscript with a great interest. Below you can find my comments to strengthen the manuscript.

Introduction

line 55: When specifying "at least", it is enough to enter one number - the lower one.

General remark: I lacked information whether the approach used by the authors was already used in the literature, or whether it is an innovative approach. The paragraph of line 54-59 is not entirely clear to me, especially the reference to the cut-offs / norms / recommendations. Could the authors explain this further?

The aim of the work should be more emphasized, especially the innovative element of the research. 

Materials and methods:

The idea of the statistical model used in the study should be better described. I miss a reference to literature.

Results:

Table 1: For what purpose were the data for the estimated population shown and calculated? It was not described in the methodology?

Table 3: I do not understand why the authors combined into one group indyviduals with underweight and normal body weight. Especially if they simultaneously distinguished overweight and obesity. Weight deficiency may be associated with disturbed nutritional status and health, as well as may result from incorrect eating habits. I consider that these criteria should be considered separately.

Discussion:

This part is well-led and informative. I would suggest to separate the conclusions from the discussion for the convinience of the reader.

Author Response

Thank you for allowing us to submit a new draft of the manuscript. We are grateful for your insightful comments on the paper. We have been able to incorporate changes to reflect the suggestions provided (highlighted in yellow). We hope you will consider the revised manuscript acceptable for publication.

Below, we provide a point-by-point response to the reviewer’s comments and concerns.

1. Introduction

line 55: When specifying "at least", it is enough to enter one number - the lower one.

Authors: The sentence was revised, accordingly.

2. General remark: I lacked information whether the approach used by the authors was already used in the literature, or whether it is an innovative approach. The paragraph of line 54-59 is not entirely clear to me, especially the reference to the cut-offs / norms / recommendations. Could the authors explain this further?

Authors: Seldom, latent class analysis has been used to model dietary transitions, but dietary intake data come from FFQs which do not have the problem of within-person variability. With 24-hours recalls and food records we need a methodological approach to overcome this. In this study, we used a latent class transition model to calculate the usual prevalence of dietary patterns (i.e. the prevalence at the long-run, minimizing the within-person variability of dietary intake) based on a transition probability matrix. The usual procedure (in previous studies) is to average the food weight intake, but this does not resolve the problem of within-person variability, and thus the intake is biased.

We change text to clarify this (page 2, lines 52-60).

To our knowledge there is no studies applying the latent class transition model to data from FFQs.

3. The aim of the work should be more emphasized, especially the innovative element of the research.

Authors: The innovative nature of the methods was highlighted in the objective description (lines 61-2).

4. Materials and methods:

The idea of the statistical model used in the study should be better described. I miss a reference to literature.

Authors: We have now included some references of studies applying the latent transition modelling to dietary data in the Introduction section (line 53). We have also included a key reference of the statistical method (line 222).

5. Results:

Table 1: For what purpose were the data for the estimated population shown and calculated? It was not described in the methodology?

Authors: The study sample was selected to represent the adult Portuguese general population. As described in the methods (page 5, lines 211-214) all prevalence estimates, and the respective 95% confidence intervals (95%CI), were weighted according to the complex sampling design. Thus, for example the 1411 subjects of the study sample belonging to dietary pattern 1 represent 3 632 681 subjects of the Portuguese population.  

6. Table 3: I do not understand why the authors combined into one group individuals with underweight and normal body weight. Especially if they simultaneously distinguished overweight and obesity. Weight deficiency may be associated with disturbed nutritional status and health, as well as may result from incorrect eating habits. I consider that these criteria should be considered separately.

Authors: Thank you for your suggestion, but due to the very low prevalence of underweight individuals (1%), as previously published (Oliveira A. BMC Public Health (2018) 18:614), this category was merged with the normal weight group. For statistical purposes, we do not have power to put them alone in a single category.

7. Discussion:

This part is well-led and informative. I would suggest to separate the conclusions from the discussion for the convenience of the reader.

Authors: Thank you for the suggestion. The conclusions are now in a separate section, as suggested.

Reviewer 3 Report

Nutrients    Manuscript ID: nutrients-1033097

The study under review aims to derive habitual dietary patterns of the Portuguese adult population of the

National Food, Nutrition and Physical Activity Survey by applying two methodological approaches: a latent class model and a latent transition model. 

The authors used the latent transition model and identified 22 patterns.  Three age and sex,-related dietary patterns were identified: “Diet-in-transition” 30 (48%), “Western” (36%) and “Traditional-Healthier” (16%),  Most of the participants followed the same pattern in both days, but 26% followed both Diet in transition and Western Diet patterns in the two days.

Comments

The paper is well written and contains novel information.

This reviewer has a few comments:

  1. The authors should report and comment the validity and interindividual figures of variability of the two-days recall diet recording, especially  in comparison with the food frequency questionnaires.
  1. Is the “Diet-in-transition” in between “Traditional-Healthier” and “Western”? could you try to find a name that suggest which kind of transition is?
  1. Traditional- Healthier is identical/similar of Atlantic Diet ?
  1. I do not understand the meaning of the following sentence in Results:

Lines 367-369:  In the current study, individuals following the “Diet-in-transition” have the highest intake of red meat and processed meat, starchy foods (pasta/rice/potatoes, bread), alcoholic beverages, particularly wine, but also sweets, soft drinks and coffee.     The follower of the diet-in-transition had the highest intake of meat and etc. of all DP?  Including  the Western DP?  But then in the following sentence the authors state  that Western diet ‘ is characterized by those having the highest intake of vegetables, protein foods, such as all type of meats and eggs, starchy foods (pasta/rice/potatoes), salty snacks, sweets, nectars/soft drinks, and water .

I believe that there is the need of a better definition of dietary patterns

  1. Results indicate that ( 305- on): Followers of the “Diet-in-transition” are less educated, and more often under/normal weighted. The “Western” DP is more often reported by more educated subjects, more frequently obese, less 306 physically active, but also with lower prevalence of diseases needing medical care. The “Traditional- Healthier” followers are more educated and physically active but have higher prevalence of pre-obesity and obesity and more diseases needing medical care.   

However, there is no mention of those results in Discussion, nor the implication they may have in the educational programs.

Author Response

Thank you for allowing us to submit a new draft of the manuscript. We are grateful to the reviewer for the insightful comments on the paper. We have been able to incorporate changes to reflect the suggestions provided (highlighted in yellow in themanuscript). We hope you will consider the revised manuscript acceptable for publication.

Below, we provide a point-by-point response to the reviewers’ comments and concerns.

Comments

 The paper is well written and contains novel information. This reviewer has a few comments:

  1. The authors should report and comment the validity and interindividual figures of variability of the two-days recall diet recording, especially in comparison with the food frequency questionnaires.

 Authors: In this study, we do not have a full FFQ to assess dietary intake and to compare with data from the two 24-hours recalls; we only have a short FFQ for fruit and vegetables (F&V) for the same participants. For these food items, we correlated data from the 2x 24h-recalls with data from the FFQ, and moderate Pearson correlation coefficients were found (fruits: rho=0.54, 95%CI: 0.52-0.56; vegetables: rho=0.33, 95%CI: 0.30-0.36; vegetable soup: rho=0.41, 95%CI: 0.38-0.44), supporting the “validity” of the 24-hour recalls method.

In a previous study using data from the same Survey, the accuracy of the eAT24 software to assess dietary intake was also measured against urinary biomarkers: N (nitrogen), K (potassium) and Na (sodium) in a subsample of 94 subjects. The eAT24 performed well in estimating protein and K intakes, but lesser so in estimating Na intake, using two non-consecutive 24-hour recalls (Goios, et al. Public Health Nutr 2020 Dec;23(17):3093-3103).

We added all this information to text (lines 144-150).

In Supplementary Material, Table S1, we also present the intra-class correlation coefficient (ICC) for each food group, measuring the inter-person variability of dietary intake of each food group.

2. Is the “Diet-in-transition” in between “Traditional-Healthier” and “Western”? could you try to find a name that suggest which kind of transition is?

 Authors: The names attributed to each pattern are arbitrary and prone to discussion. The “Diet-in-transition” dietary pattern is characterised by the highest intake of legumes, starchy foods, such as pasta, rice, potatoes and bread, red and processed meats, and wine (considered as our traditional Atlantic diet), but it also includes sweets, sugar, soft drinks and coffee; so, we believe that it represents a transition between more traditional-healthy habits and westernized food habits. For that reason, we suggest to call it as “In-transition to Western”, clarifying in that way for where we are moving forward.

3. Traditional- Healthier is identical/similar of Atlantic Diet ?

Authors: As suggested in our discussion (page x12, lines 436-439), we believe that according to our data, the “Traditional-Healthier” dietary pattern is more close to the Mediterranean Diet as it is characterized by the highest intake of foods, such as vegetables, including the vegetable soup, fruits, dairy, fishery, and olive oil. It does not include a high intake of meats and alcoholic beverages, for example. On the other hand, individuals following the “Diet-in-transition” have the highest intake of red meat and processed meat, starchy foods (pasta/rice/potatoes, bread), alcoholic beverages, particularly wine. Thus, we hypothesised that this dietary pattern has more similarities with the Atlantic Diet that has been defined for the first time in the literature by our research group (Oliveira A. Am J Clin Nutr 2010, 92, 211-7) as the traditional diet from Northern Portugal and Galiza, Spain.

4. I do not understand the meaning of the following sentence in Results:

Lines 367-369:  In the current study, individuals following the “Diet-in-transition” have the highest intake of red meat and processed meat, starchy foods (pasta/rice/potatoes, bread), alcoholic beverages, particularly wine, but also sweets, soft drinks and coffee.     The follower of the diet-in-transition had the highest intake of meat and etc. of all DP?  Including  the Western DP?  But then in the following sentence the authors state  that Western diet ‘ is characterized by those having the highest intake of vegetables, protein foods, such as all type of meats and eggs, starchy foods (pasta/rice/potatoes), salty snacks, sweets, nectars/soft drinks, and water .

I believe that there is the need of a better definition of dietary patterns

Authors: The dietary pattern definition is based on the distribution of individuals across 3 classes of food consumption: nil; equal or below the sample median and above the sample median intake. To define which food groups are more representative of each dietary pattern we should look for the percentages in each class (Table 2). Sometimes, they are very close, and for that reason a food group could be “representative" for more than one dietary pattern.

Nevertheless, the correct way of describing this is to say that the “In-transition to Western” has a high percentage of individuals with the highest intake (in the upper category of intake) of red meat and processed meat, starchy foods (pasta/rice/potatoes, bread), alcoholic beverages, particularly wine, but also sweets, soft drinks and coffee. The text was revised, accordingly.

5. Results indicate that ( 305- on): Followers of the “Diet-in-transition” are less educated, and more often under/normal weighted. The “Western” DP is more often reported by more educated subjects, more frequently obese, less 306 physically active, but also with lower prevalence of diseases needing medical care. The “Traditional- Healthier” followers are more educated and physically active but have higher prevalence of pre-obesity and obesity and more diseases needing medical care.  However, there is no mention of those results in Discussion, nor the implication they may have in the educational programs.

Authors: To study the association of dietary patterns identified according to nutritional intake, geographical region, socioeconomic and health status was not the primary aim of our study. As discussed, we highlighted a possible reverse causality bias, due to the cross-sectional nature of this study, and for that reason, these results were not extensively discussed. They had only descriptive purposes. Nonetheless, we discussed the differences in age and sex of the membership in each dietary pattern: “Diet-in-transition” is more frequently followed by middle-aged men; the “Traditional-Healthier” is more frequently reported by older women; and the “Western” is followed by younger individuals, and the implication of a possible cohort effect rather than an age effect be present in our results. This means that it is not expected that the population change their food habits throughout life, i.e. that older individuals will adopt healthier dietary patterns; it is rather expected that in a few years dietary patterns will worsen with health-related implications.

Round 2

Reviewer 2 Report

Dear Authors,

Thank you for responding my comments.

Author Response

Thank you for your feedback.

Reviewer 3 Report

Point 4.  the authors should add the explanation provided in the text. 

Point 5.  The authors state that 'socioeconomic and health status was not the primary aim of our study'.   

and... 'This means that it is not expected that the population change their food habits throughout life, i.e. that older individuals will adopt healthier dietary patterns; it is rather expected that in a few years dietary patterns will worsen with health-related implications'

In general, there is no point in planning a study without expecting that it could be useful in promoting health. 

If the Authors present socioeconomic and health state in results, they should appropriately discuss and speculate about them in Discussion. It is not appropriate that some sort of discussion about the findings is done in the Results Section.   

I believe that the Authors must address point 5 with appropriate discussion of the socioeconomic findings and educational derivates.    However, if the Authors believe that education is not useful and people in the future will only worsen their dietary habits, then they must support their belief with appropriate references.  

Author Response

Thank you for allowing us to submit a new draft of the manuscript titled: ‘Application of a latent transition model to estimate the usual prevalence of dietary patterns’ to Nutrients. 

A point-by-point response to the reviewers’ comments and concerns is provided below, and changes are highlighted in yellow in the manuscript.

Point 4.  the authors should add the explanation provided in the text. 

Authors: The explanation was added to the methods section (please see page 4, lines 155-60):

“The decision on the food groups inclusion in each dietary pattern was based on the distribution of individuals across these three classes of food consumption: nil; equal or below the sample median and above the sample median intake. To define which food groups are more representative of each dietary pattern, the percentages of individuals in each class were observed. If they are very close, a food group could be “representative" for more than one dietary pattern.”  

Point 5.  The authors state that 'socioeconomic and health status was not the primary aim of our study'.   and... 'This means that it is not expected that the population change their food habits throughout life, i.e. that older individuals will adopt healthier dietary patterns; it is rather expected that in a few years dietary patterns will worsen with health-related implications'

In general, there is no point in planning a study without expecting that it could be useful in promoting health. 

If the Authors present socioeconomic and health state in results, they should appropriately discuss and speculate about them in Discussion. It is not appropriate that some sort of discussion about the findings is done in the Results Section.   

I believe that the Authors must address point 5 with appropriate discussion of the socioeconomic findings and educational derivates. However, if the Authors believe that education is not useful and people in the future will only worsen their dietary habits, then they must support their belief with appropriate references.  

Authors: Previous evidence supports the relationship between higher socio-economic status and healthier dietary patterns, as shown in our study. The followers of the ”Traditional-Healthier” dietary pattern  are the most educated (after standardization for age and sex across dietary patterns). From a public health perspective, we cannot change the educational level of an entire population, but we can focus on more socio-economic (vulnerable) target groups of the population for health promotion strategies.

As we previously stated, the cross-sectional nature of a Survey is not the appropriate design to identify factors influencing the development of specific dietary patterns and thus target groups for intervention. Nonetheless, we now discussed that socio-economic status was positively associated with healthier dietary patterns, as supported by previous evidence.

Please see page 12, lines 412-19 to see:

“Also, a higher educational level was associated with the “Traditional-Healthier” DP. As previously suggested, a healthier diet in individuals from high socio-economic status is one of the mechanisms that explains social differences in health [29]. Indeed, in high-income countries, high-socioeconomic status individuals are more likely to consume healthy foods such as whole grains, lean meats, fish, and fruit and vegetables, whereas more disadvantage socioeconomic individuals tend to consume more fat and less fiber [29, 30]. Thus, younger and less educated subjects should have a special focus when designing health promotion strategies related with diet.”

 We also added two references supporting the tracking of dietary patterns [29, 30].

  1. Movassagh, E.Z., Baxter-Jones, A.D.G., Kontulainen, S., Whiting, S.J., Vatanparast, H. Tracking Dietary Patterns over 20 Years from Childhood through Adolescence into Young Adulthood: The Saskatchewan Pediatric Bone Mineral Accrual Study. Nutrients 2017, 9,
  1. Ambrosini, G.L., Emmett, P.M., Northstone, K., Jebb, S.A. Tracking a dietary pattern associated with increased adiposity in childhood and adolescence. Obesity (Silver Spring) 2014, 22, 458-65.